# Identifying an Optimal Model for Blockchain Technology Adoption in the Agricultural Sector

Peter Sasitharan Gandhi Maniam *, Catherine Prentice, Anne-Marie Sassenberg and Jeffrey Soar

School of Business, University of Southern Queensland, Toowoomba, QLD 4350, Australia;
cathyjournalarticles@gmail.com (C.P.); anne-marie.sassenberg@usq.edu.au (A.-M.S.); jeffrey.soar@usq.edu.au (J.S.)
* Correspondence: peter.gandhimaniam@usq.edu.au

**Abstract:** *Background*: Blockchain adoption in agriculture is facing challenges. Some of its key challenges include lack of regulatory framework and unclear policies. *Methods*: This quantitative research based on a survey aims to examine the factors that influence blockchain implementation in the agricultural sector. Several theories including the technology acceptance model (TAM), the theory of planned behaviour (TPB), and the technology readiness index (TRI) were drawn upon to determine the factors influencing Blockchain adoption. The study was undertaken in Australia with 358 agricultural supply chain experts. *Results*: A range of novel findings were generated. While perceived usefulness, perceived ease of use, and attitude positively influenced Blockchain adoption, discomfort and insecurity were considered deterrents. *Conclusions*: The results can be used by relevant practitioners to improve the supply chain management for agriculture business. The findings can also inform a new direction for the research on the agricultural supply chain and the literature on logistics.

**Keywords:** blockchain adoption; agriculture; supply chain management





## 1. Introduction

Blockchain technology (BCT) has emerged as one of the technological solutions to enhance the coordination, collaboration, and traceability of supply chain transactions [1–3]. BCT has several potential advantages over traditional supply chain management including being tamper-free, reducing costs and losses, enhancing trust, and offering decentralised information [2]. Blockchain architecture offers greater transparency and traceability of supply chain transactions, effectively reducing any trust problems among various parties [4].

There is an increase in the BCT uptake in the food and agricultural supply chain [5]. Kamilaris et al. [5] indicate that BCT is a promising technology for creating transparency and trust within agricultural processing and manufacturing and for reducing the cost of operations in manufacturing. BCT adoption in the agricultural supply chain is advantageous in supply chain traceability [6], transparency, financial returns [7], fresh produce supplies [8], and food supply and logistics management [1]. BCT has been found to contribute to improved food supply chain effectiveness [9] and the enhanced traceability of food produce distribution [2,10]. Implementing BCT in agricultural supply chains may contribute to reduced redundancy, shorter lead times, a leaner supply chain, and fewer delays [8]. BCT adoption in supply chains ensures high-quality standards, giving stakeholders more control of the production and distribution of agricultural produce across the supply chain [7]. It can also contribute to improved safety, privacy, and individual control of data in the food processing supply chain industries [8,11]. Despite these reported advantages, the adoption of BCT has not received sufficient attention in the agricultural sectors [8]. Some of the potential challenges include unclear policies; inadequate regulatory framework, education, and technical aspects [5,12]; a high implementation cost [8,13]; perceived risks; and a lack of relevant knowledge [1].

While much remains to be known about supply chain adoption in organisations, very few studies have attempted to examine the factors influencing the adoption of BCT

in agricultural supply chains [3]. The current study seeks to address this gap. Three theoretical models including the technology acceptance model (TAM), the theory of planned behaviour (TPB), and the technology readiness index (TRI) are drawn up to identify the optimal factors in adopting BCT in the agricultural sector. TAM, TPB, and TRI offer a comprehensive understanding of the factors influencing BCT adoption in the agricultural supply chain [14]. Both the PEU and PU are cognitive dimensions that predict individual technology acceptance and are related to TRI constructs based on behavioural intentions and individual psychological differences [15,16]. Perceived behaviour and subjective norms in the TPB model are used to understand how control influences users to adopt technology when combined with TAM constructs [15].

The main objective of this study is to estimate an optimal model that identifies the key factors that influence BCT implementation in the Australian agricultural sector. Using a quantitative approach, this study examines the antecedents of BCT adoption within the context of the Australian agricultural sector. The motivations for this research study are twofold. First, there is an inspiration to address challenges related to the low BCT uptake in the Australian agricultural sector using an optimal quantitative model. The findings are expected to highlight the drivers and inhibitors to BCT implementation in the agricultural sector. The effective management of these factors (i.e., antecedents) is expected to address the stated challenge in BCT adoption. Second, the study draws motivation to extend the existing theoretical insight/knowledge on BCT implementation using prominent technology-related theories such as TAM, TPB, and TRI. It is expected that the integration of the findings with the three technology-related models/theories will identify key factors or constructs that influence BCT implementation in the Australian agricultural sector.

This study contributes to the research on supply chain management and may provide measures to enhance the adoption of BCT in the agricultural sector. The following section discusses the relevant literature and offers hypotheses, and then outlines the methodology for testing the proposed relationships. The results of this study are presented, while the discussion and implications of this study's findings conclude this paper.

*1.1. Literature Review and Hypothesis Development*

1.1.1. Blockchain Technology

BCT is a decentralised and distributed ledger that records digital assets [17] and information in a way that makes it difficult to hack or alter [18]. BCT has also been defined as a network of computers that operate without the need for a central authority [6,10]. BCT is decentralised, difficult to maliciously manipulate, and can therefore help to ensure the trustworthiness of multiple entities in a transaction. Being a peer-to-peer transaction platform, BCT does not require third parties, as different entities involved in a transaction serve as nodes, with all transactions varied using cryptography [8]. BCT can help to eliminate trust-related problems in business transactions, thereby reducing the friction of business authentication [19] contributing to transparency in financing transactions, and enhancing the traceability of supply chains [7,8].

Although most of the BCT uptake has been limited to computer science and finance, there is an increased utilisation of blockchain in other fields like livestock, agriculture, the environmental field, and agribusiness logistics [20]. BCT adoption in agribusiness has been accelerated by several factors, such as the changing consumer demand, the need for accountability, traceability, sustainability, and the need to address the challenge of farm produce perishability [21]. This technology can bridge the trust gap between producers and consumers, where the former can inform customers on the source and processing journey of the food they consume. However, the scale of adopting BCT in the agricultural sector is rather limited.

Technology Acceptance Model (TAM)

The TAM framework was proposed by Davis in 1989 to explain how individuals accept and use technology. Two primary factors that influence the use of technology by

potential users are perceived usefulness (PU) and the perceived ease of use (PEU) [22]. The primary feature under the TAM is an emphasis on user perception. Davis first used the TAM in 1989 to explain the key determinants of technology acceptance. Figure 1 shows the two constructs of perceived usefulness and perceived ease of use initially proposed by Davis [22]. Perceived usefulness refers to the subjective likelihood that using technology will improve users' actions. Perceived ease of use refers to the likelihood that using a system will be effortless [22].

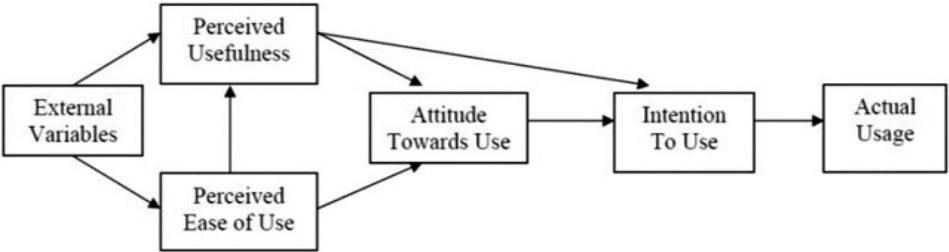

**Figure 1.** First modified technology acceptance model (TAM) [22].

In 1996, Venkatesh and Davis improved on the initial TAM after finding that both the PU and PEU directly influenced behavioural intention [23]. The new TAM eliminated the attitude construct. Figure 2 shows the final version of the TAM.

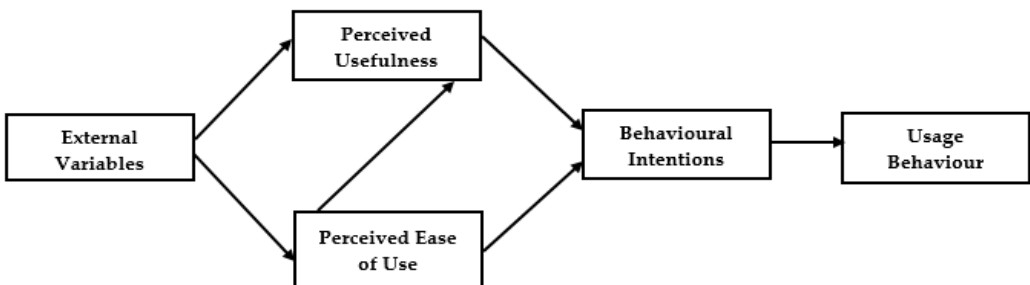

**Figure 2.** The final version of the TAM. Adapted with permission [23].

Table 1 presents examples of some past empirical studies that have used the TAM adoption model to understand the acceptance of using BCT in supply chains. Past studies investigating the perceived usefulness and perceived ease of use focus on private and public supply chains [24], logistics [25,26] and the grape wine supply chain Blockchain is an emerging technology that has largely been limited to the information technology and financial sectors [4]. Being a novel technology and considering the diversity of the agricultural sector, the factors that may influence its uptake in this industry may differ from other sectors. The TAM has been widely used in recent studies to assess how perceived usefulness and perceived ease of use inform BCT uptake in agricultural supply chains, and also to examine the user behaviour and acceptance of BCT in agricultural supply chains [4]. For instance, the TAM has been applied to assess the BCT impact in facilitating an accurate, secure, real-time, and cost-effective coffee supply chain in Burundi, with implications on the need for further research on its application in agricultural supply chains.

**Table 1.** Past empirical studies that have used TAM adoption models to explore blockchain acceptance in supply chains.

| References | Country | Objectives | Findings |
|---|---|---|---|
| Giri and Manohar [24]. | India | To examine the acceptance of private and public BCT-based collaboration among supply chain practitioners | Collaboration strongly mediated the relationship between both perceived usefulness and perceived ease of use and their influence on behavioural intention to use |
| Jain et al. [25]. | India | To understand blockchain uptake and acceptance in logistics | Perceived usefulness, perceived ease of use, and attitude influence BCT uptake and implementation in logistic supply chains |
| Queiroz and Fosso Wamba [26]. | United States and India | To understand BCT adoption behaviour in the logistics and supply chain fields in India and the USA | Supply chain and logistic transactions executed using blockchain were deemed to be safer, more traceable, and transparent |
| Saurabh and Dey [4]. | India | To identify potential factors of BCT adoption in the grape wine supply chain | Trust, compliance, traceability, dis-intermediation, control, and coordination inform BCT adoption |
| Shrestha and Vassileva [27]. | Canada | To uncover how the PEU, perceived enjoyment, system quality, and perceived usability influence the intention to use BCT-based systems | Behaviour influences intention to use BCT in the supply chain, with the quality of the system having a strong influence on perceived usefulness and perceived ease of using the technology |

Theory of Planned Behaviour (TPB)

Ajzen [28] proposed the TPB to help understand and explain how individual behaviours are influenced by intention, and how intentions are related to perceived behavioural control, subjective norms, and attitude towards behaviour. Figure 3 presents the TPB. The first two constructs (i.e., attitude and subjective norms) are like the theory of reasonable action proposed by Fishbein and Ajzen in 1975 [29]. The third construct (i.e., perceived behaviour control) is the limit that users consider may hinder their behaviour. The TPB is important when modelling the acceptance of various new information technology products and assessing levels of usage [30].

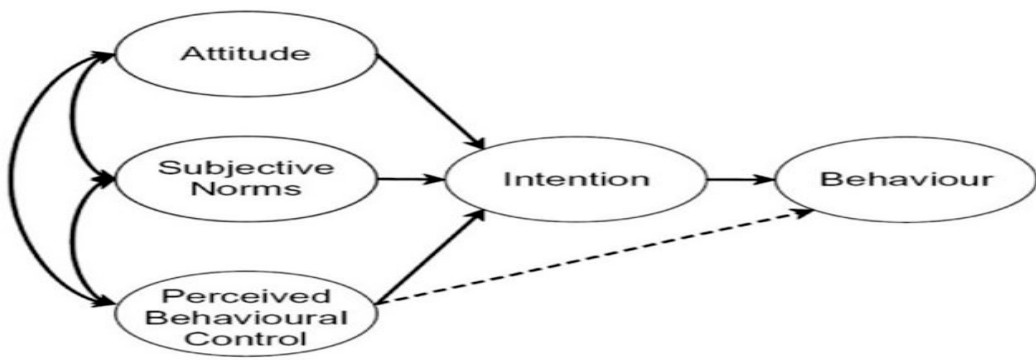

**Figure 3.** The theory of planned behaviour. Adapted with permission [28].

The TPB includes subjective norms and perceived behavioural control. Subjective norms indicate a person's perception about what significant others think they should consider when adopting new technology [17]. Subjective norms influence behavioural intentions among organisational leaders, while other researchers have observed that subjective norms influence perceived usefulness among managers to use BCT [1,2]. Further research was undertaken to improve the TRA to form the TPB by adding perceived behaviour control, which measures users' behavioural intentions [28,29]. Perceived behavioural control is an individual perception about one's personal abilities to perform a specific activity [2]. Earlier research evidence on BCT adoption was largely a theoretical investigation focusing on supply chain systems in the agricultural sector [2,20].

Technology Readiness Index (TRI)

The TRI measures individual readiness to use technology. The TRI is defined as "people's propensity to embrace and use new technologies for accomplishing goals in home life and at work" [2] (p. 308). The TRI is used to understand people's beliefs and it contains four sub-constructs: insecurity, discomfort, innovativeness, and optimism [31,32]. Optimism defines the positive perception a user has towards technology belief for improved efficiency, flexible operations, and control [33,34]. Innovativeness means a sense of inclination and a belief that a person has been a pioneer with newly introduced technology [33]. Discomfort refers to a sense of being overwhelmed and lacking control when using new technology [33,35]. Finally, insecurity refers to distrusts and worries about using new technology, and potential users largely remain suspicious of its capabilities to be helpful in their work or at home [33].

Over the years, researchers have used the TRI to understand technology use, where innovativeness and optimism are considered motives of use, while discomfort and insecurity are considered inhibitors [4,14,36]. The TRI has been used in combination with the TAM to assess blockchain use in enhancing horticulture traceability [37] and establishing fair agricultural trade [2]. Table 2 presents a review of the literature on the use of BCT in the agricultural supply chain management from various countries, showing that there is a paucity of research on the topic as in the Australian context.

**Table 2.** Past empirical studies on blockchain use in agricultural supply chain management.

| References | Country | Objectives | Findings |
|---|---|---|---|
| Ali et al. [37] | Malaysia | To propose an effective blockchain framework to enhance the integrity of the halal food supply chain | BCT uptake in the halal supply chain is influenced by regulatory capability, efficient production, change management, cost reduction, and logistic efficiency |
| Bischoff and Seuring [6] | Germany | To identify limitations and opportunities that influence the adoption of BCT in supply chain management | Blockchain adoption acceptance is influenced by information confidentiality, the privacy of entities in a supply chain, and vulnerability towards third parties |
| Collart and Canales [8] | United States | To assess the adoption of BCT and its potential impact in addressing the challenges of the fresh produce industry | BCT adoption enhances the resilience of supply chains by reducing food fraud, loss, wastage, and ensuring safety |
| Hu et al. [38] | China | To formulate a blockchain framework to enhance efficiency and reduce cost in the organic supply chain | BCT use in supply chains may contribute to enhanced efficiency, cost effectiveness, transparency, tamper resistance, and trust-free and immutable constructs |
| Kamble et al. [2] | India | To identify and establish the relationship between the enablers of BCT adoption in agricultural supply chains | Traceability, auditability, provenance, and immutability largely inform the uptake and use of BCT in Indian agricultural supply chains |

### 1.2. TRI, TAM, and TPB

The TRI constructs include discomfort and insecurity. BCT adoption may be impacted by inhibitors, with the key among them being discomfort [33,39]. Inhibitors potentially affect the technology readiness of managers in organisations. Discomfort is defined as a perceived lack of control over technology and a feeling of being overwhelmed by innovations [33]. Based on perceived behavioural control, it may be anticipated that the relationship between discomfort and BCT adoption would be negative. The TPB suggests that perceived behavioural control is a direct determinant of both actual behaviour and behavioural intention [29]. Previous findings show that discomfort (a users' general feeling of lack of control) should have a negative effect on BCT uptake. Users who have high discomfort levels towards new technology find it less easy to use it [40]. The current study differs from previous research by undertaking an empirical assessment to determine the effect of discomfort on perceived usefulness, perceived ease of use, attitude towards use, and perceived behavioural control when adopting BCT in agricultural supply chains. Discomfort has been found to negatively influence the perceived usefulness and the perceived ease of use of BCT as it inhibits its adoption as a new technology among agricultural supply chain managers [18,41]. Furthermore, discomfort tends to negatively influence the attitude towards use and the perceived behavioural control of new technology [41]. Consequently, the following hypotheses are postulated:

**Hypothesis 1 (H1).** *Discomfort (DISC) with blockchain negatively affects the perceived ease of use (PEU) of BCT.*

**Hypothesis 2 (H2).** *Discomfort (DISC) with blockchain negatively affects the perceived usefulness (PU) of BCT.*

**Hypothesis 3 (H3).** *Discomfort (DISC) with blockchain negatively affects attitudes towards the use (ATT) of BCT.*

**Hypothesis 4 (H4).** *Discomfort (DISC) while using blockchain negatively affects the perceived behavioural control (PBC) of BCT.*

Insecurity denotes an individual's level of distrust in a new technology. Distrust may stem from scepticism regarding its capacity to work properly or personal concerns about possible harmful consequences [33,42]. Insecurity is combined with general safety issues, apprehensions about negative consequences, and the desire for assurance [17,43]. In organisations, if managers are naturally distrustful of and sceptical about technology, they are likely to anticipate risks instead of benefits from the implementation of technology [2]. As a result, individuals are likely to avoid its uptake. In line with the TPB, one would expect a negative relationship between the insecurity trait and technology usage. Past studies have not examined how insecurity might influence individual behaviour towards BCT adoption, thereby showing the need for this study. The results of this study will create new knowledge regarding insecurity as a potential technology readiness inhibitor hindering the usage intention and usage behaviour of BCT in agricultural supply chains.

Insecurity may contribute to the low utilisation of BCT in agricultural sectors in addition to ambiguity [41]. Insecurity inhibits the individual uptake of BCT in agricultural supply chains [10]. Insecure managers are less likely to embrace BCT uptake as they express less support on whether its use will be beneficial in facilitating efficient supply chains [10]. Insecurity may contribute to a low perceived usefulness and a low perceived ease of use of a technology, potentially hindering its uptake within organisations [1,9,38]. Insecurity could also negatively influence perceived behavioural control and attitude towards the use of BCT [38]. Therefore, based on the stated prior research evidence, the following is postulated:

**Hypothesis 5 (H5).** *Insecurity (INSC) negatively affects the perceived ease of use (PEU) of BCT.*

**Hypothesis 6 (H6).** *Insecurity (INSC) negatively affects the perceived usefulness (PU) of BCT.*

**Hypothesis 7 (H7).** *Insecurity (INSC) negatively affects the perceived behavioural control (PBC) of BCT.*

**Hypothesis 8 (H8).** *Insecurity (INSC) negatively affects the attitude towards use (ATT) of BCT.*

A growing body of research has shown that PEU substantially impacts managers' usage intention when considering BCT technology in agricultural supply chains [8,44]. PEU denotes the degree to which managers in agricultural organisations believe that using BCT would improve their supply chain management and transparency [7]. The implication is that the perceived ease of use of a novel technology would influence managers' intentions to implement the technology within the organisation [7]. Consistent with the foregoing discussion, the following hypothesis is offered:

**Hypothesis 9 (H9).** *Perceived ease of use (PEU) positively affects the intentions to use BCT.*

Perceived usefulness refers to the degree to which managers in agricultural organisations believe that using BCT would improve their logistics and supply chain process performance [6]. For example, corporations that find BCT reliable in ensuring the effective supply and delivery of halal food are likely to associate the technology with greater usefulness [40]. Recent studies show that perceived usefulness influences managers' intentions to use BCT in supply chains [2,43]. Hence, the study offers the following hypothesis:

**Hypothesis 10 (H10).** *Perceived usefulness positively affects the intention to use BCT.*

Further research was undertaken to improve the TRA to form the TPB by adding perceived behaviour control, which measures users' behavioural intentions [28,29]. Perceived behavioural control is an individual perception about one's personal abilities to perform a specific activity [2]. Perceived behavioural control has been noted to have a positive influence on behavioural intentions to use BCT in the agricultural sector [2]. Therefore, the study proposes the following hypothesis:

**Hypothesis 11 (H11).** *Perceived behavioural control positively affects the intention to use BCT.*

Attitude largely captures an emotional aspect of managers' intentions to use BCT in their organisations when seeking to improve their supply chain management [3,17]. Researchers report that attitude largely captures the emotional aspect of users' intention to use new technology. Attitude defines the level to which users show a favourable or unfavourable assessment of technology [41]. Earlier research evidence indicates that positive attitude influences managers' behavioural intentions to use BCT [41]. Consequently, this study proposes the following hypothesis:

**Hypothesis 12 (H12).** *Attitude positively affects the intention to use BCT.*

Figure 4 presents the proposed theoretical model for this study.

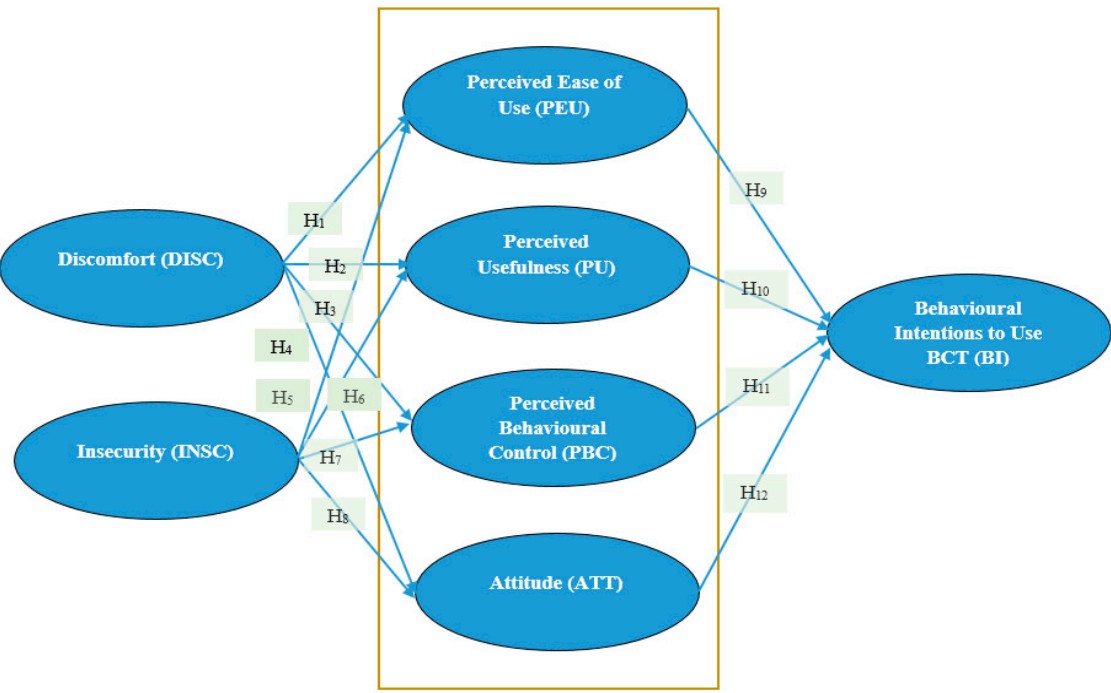

**Figure 4.** Proposed theoretical framework.

## 2. Methods

### 2.1. Sample

Quantitative data was collected through an online survey based on a sample of 385 professionals drawn from the Australian agricultural industry. The criterion for selection is that they had to be employees or managers of firms that operate in the Australian agricultural sector. The other criterion for selection is that their respective firms must have deployed BCT in their supply chains. The stakeholders also need to have experience in how BCT operates in their supply chains. Purposive sampling, a nonprobability sampling technique, was used to recruit participants to the study. The focus on purposive sampling was informed by the need to recruit a sample with expertise and relevant information on blockchain adoption in agricultural supply chains [45]. There are multiple entities in the Australian supply chain including vendors, producers, warehouses, transportation companies, retailers, and distribution centres [46].

### 2.2. Measurement

This study employed existing measures to measure the study constructs of the proposed theoretical framework model with a 5-point Likert scale, which ranged from 1 = "Strongly Disagree" to 5 = "Strongly Agree". Discomfort was measured using four items that were adapted from Godoe [39] and Parasuraman [33]. The four dimensions of discomfort were understandability, complexity, knowledgeable, and reliability. The Cronbach's alpha values for the four sub-factors were 0.76 (understandability), 0.71 (complexity), 0.83 (knowledgeable), and 0.78 (reliability). The Cronbach's alpha value for the discomfort scale was 0.80.

The insecurity scale was measured using six items that were adapted from Parasuraman [42] and Parasuraman [33]. The six sub-scales that were used to measure the insecurity construct were safety, privacy, accessibility, confirmation, accuracy, and interaction. All six sub-scales had fairly high internal reliability scores as follows: 0.72 (safety), 0.75 (privacy), 0.77 (accessibility), 0.74 (confirmation), 0.93 (accuracy), and 0.91 (interaction). The overall Cronbach's alpha value for the insecurity construct was 0.79.

The measure for the perceived usefulness scale was adapted from Davis, Bagozzi, and Warshaw [22] and Godoe [39]. There were four dimensions or items that were used to

measure the perceived usefulness scale. These include timeliness, performance, productivity, and effectiveness. The four dimensions had high internal reliability values as follows: 0.81 (timeliness), 0.92 (performance), 0.95 (productivity), and 0.88 (effectiveness). The estimated Cronbach's alpha value for the perceived usefulness scale was found to be 0.90.

The perceived ease of use was also assessed using the Davis, Bagozzi, and Warshaw [22] and Godoe [39] measures. The four perceived ease of use dimensions were usability, understandability, recall, and efficacy. The internal reliabilities for each of the four sub-factors were 0.73 (usability), 0.81 (understandability), 0.88 (recall), and 0.91 (efficacy). The perceived ease of use scale had a fairly high internal reliability given its Cronbach's alpha value of 0.76.

The five items that were used to measure the attitude towards technology were adapted from Davis et al. [22] and Godoe [39]. The five dimensions with their corresponding internal reliabilities were desirability (0.81), applicability (0.89), benefit (0.77), favourability (0.77), and adoption (0.82). Given its Cronbach's alpha value of 0.89, the overall attitude towards technology scale had a fairly internal reliability.

The perceived behavioural control scale was measured by adapting the 3-item measurements developed by Aboelmaged and Gebba [47]. The three sub-factors were confidence, control, and ability. Their internal reliability scores were 0.90, 0.88, and 0.92, respectively. The overall perceived behavioural control scale had a high internal consistency given its Cronbach's alpha value of 0.91. Finally, the three items measuring the behavioural intention scale, including intention, expectation, and adoption, were adapted from Ho and Ko [48] as well as Vankatesh and Zhang [49].

### 2.3. Data Collection Procedure

To recruit participants, the data collection process was outsourced to Zoho, a web-based survey tool, to conduct survey research, evaluations, and other data collection initiatives (https://www.zoho.com/survey/, accessed on 1 March 2021). An online advertisement was posted on Facebook targeting stakeholders across various agricultural sectors in Australia. The advertisement contained details of the study including the aim and objectives of the study, as well as a consent form from the university to conduct the study. Alongside the advertisement, detailed information about the study was included, together with a formal request to invite participants to participate in the study. Participants who expressed interest in participating in the study accessed a provided URL link where the online survey was hosted. Completed surveys were returned anonymously to conceal the identity of the participants.

The first stage of data collection consisted of an initial pilot test conducted with 10 professionals from across the agricultural supply chain to evaluate the reliability of the measurement items. The experts who participated in the pilot study were recruited based on pre-established contacts. The second stage of data collection consisted of a survey. The first section of the survey included informed consent with an option to "Exit" or "Continue". Participants who clicked "Continue" were considered to have voluntarily consented to participate in the study. The emails to participants were sent between 1 March 2021 and 30 April 2021. Two follow-up reminders were performed on 20 March and 11 April 2021. At the end of the eight-week duration, a total of 385 responses were received. In total, 27 surveys were incomplete, partially filled, or not filled and subsequently excluded from the final data analysis, as discussed in Section 4.1; only 358 survey responses were used in the final data analysis. This sample size was deemed appropriate. Agriculture remains a vital economic sector employing over 385,000 people, including 136,000 farmers who provide 93% of the domestic food supply [46]. Since it is difficult to survey all stakeholders in the agricultural sector, a suitable representative sample was identified using G*Power analysis. Assuming a population size of 385,000, at a 95% confidence level, and with a margin of error of 5%, a suitable representative sample for the study was 385 participants.

Among the valid responses, 69.3% were male and 30.7% were female. Most participants (39.1%) were aged 26–35 years, followed by 33.8% of the participants who fell

within the 36–45 age bracket, and 17.3% in the 20–25 age bracket. In terms of work experience, 29.3% had worked for between 4 and 7 years, 29.3% had worked for between 8 and 12 years, 20.7% had worked for between 1 and 3 years, and 5.9% had worked for more than 13–17 years.

*2.4. Common Method Bias*

There are several measures that were undertaken to minimise common method bias. First, the online survey was designed with simple and concise questions, which reduced any potential response bias that might have occurred due to a lack of question clarity. The survey questions were formulated using only familiar terms and syntax. Furthermore, the survey was reviewed to ensure that there were no leading questions that might introduce response bias. Normality tests were performed to assess whether the data were drawn from a normally distributed population. The collected data from the Australian agricultural sectors were normally distributed and did not deviate significantly from a normal distribution. The diagnosis of multicollinearity also indicated that the response data did not have any issues of multicollinearity among the independent variables of interest.

## 3. Results

*3.1. Measurement Model*

The constructs were examined for their convergent validity. The Smart PLS software was used with reported merits to estimate the partial least squares measurement model [40,50]. All factor loadings were found to be above the recommended 0.70 level, showing a high internal consistency of the used items [50]. Further, the composite reliability (CR) for all the survey constructs exceeded the recommended limit of 0.60 [40]. Similarly, the average variance extracted (AVE) values were more than the acceptable level of 0.50 [40]. Finally, the Cronbach's alpha values for all the constructs were greater than the recommended value of 0.70, indicating an acceptable levels of scale reliability and internal consistency [45]. All the factors were used in the subsequent data analysis. The results of factor loading (FL), Cronbach's alpha coefficient, composite reliability (CR), and the average variance extracted (AVE) are presented in Table 3 below.

**Table 3.** Factor loadings for all measurement items of the study variables.

| Item | FL | Alpha | CR | AVE |
|------|----|-------|-----|-----|
| Discomfort: | | 0.80 | 0.83 | 0.64 |
| It will be difficult to understand and apply the concept of BCT in SCM. | 0.76 | | | |
| At times, BCT is thought to be designed for complex supply chain usage only. | 0.71 | | | |
| I feel that an SP who is more knowledgeable may take advantage of our SCM. | 0.83 | | | |
| Technology seems to fail at the worst possible time. | 0.78 | | | |
| Insecurity: | | 0.79 | 0.78 | 0.71 |
| I do not consider it to be safe in our firm to adopt BCT. | 0.72 | | | |
| I worry that other people will obtain the information sent via the BCT. | 0.75 | | | |
| I do not feel confident doing business on a portal that can only be reached online. | 0.77 | | | |
| Any electronic business transaction should be confirmed later in writing. | 0.74 | | | |
| For automated items, you need to check to ensure the system is error free. | 0.93 | | | |
| When you call a business, you prefer talking to a person rather than a machine. | 0.91 | | | |
| Perceived Usefulness: | | 0.90 | 0.87 | 0.80 |
| Using BCT will help minimise transaction delays. | 0.71 | | | |
| Using BCT would improve SCM performance. | 0.72 | | | |
| Using BCT would improve SCM productivity. | 0.70 | | | |
| Using BCT would improve SCM effectiveness. | 0.82 | | | |
| Perceived Ease of Use: | | 0.76 | 0.75 | 0.71 |
| The features of BCT will be easy to use. | 0.73 | | | |
| BCT is clear and understandable. | 0.95 | | | |
| It will be easy to remember and perform tasks using BCT. | 0.70 | | | |
| BCT will be easier to use compared to conventional practises of managing SCM. | 0.79 | | | |
| Attitude: | | 0.89 | 0.84 | 0.62 |
| In my opinion, it is desirable to use BCT in SCM. | 0.81 | | | |
| It will be good for SCM to use BCT. | 0.89 | | | |

**Table 3.** *Cont.*

| Item | FL | Alpha | CR | AVE |
|---|---|---|---|---|
| I guess using BCT is a good idea. | 0.77 | | | |
| Overall, I am favourable towards BCT. | 0.77 | | | |
| I will be happy if my company implements BCT. | 0.82 | | | |
| Perceived Behavioural Control: | | 0.91 | 0.87 | 0.75 |
| Our firm would be able to use BCT well. | 0.79 | | | |
| Using BT is entirely within our firm's control. | 0.88 | | | |
| Our firm has the resources, knowledge, and ability to use BCT. | 0.72 | | | |
| I foresee that our firm will use BCT regularly in the future. | 0.89 | | | |
| Our firm will use BCT in future. | 0.76 | | | |
| I expect my firm to use BCT or a similar type of system for SCM transactions. | 0.74 | | | |

Note: FL = Factor Loading, CR = Composite Reliability, AVE = Average Variance Extracted.

Table 4 indicates that discriminant validity was confirmed because the square root AVE of the examined constructs were higher than the correlations between a specific construct and other constructs in the tested model [51,52]. Discriminant validity is confirmed when the square root of AVE exceeds the absolute similarity of the construct to other constructs [51]. This requirement is satisfied as all the coefficient values in the main diagonal are greater than the non-diagonal entries.

**Table 4.** Discriminant validity and tests of differences between correlations.

| | DISC | INSC | PU | PEU | ATTI | SN | BC | BI |
|---|---|---|---|---|---|---|---|---|
| DISC | 0.670 | | | | | | | |
| INSC | 0.223 | 0.593 | | | | | | |
| PU | 0.175 | 0.546 | 0.609 | | | | | |
| PEU | 0.521 | 0.280 | 0.217 | 0.710 | | | | |
| ATT | 0.514 | 0.375 | 0.351 | 0.503 | 0.679 | | | |
| PBC | 0.546 | 0.213 | 0.225 | 0.488 | 0.574 | 0.559 | 0.639 | |
| BI | 0.454 | 0.278 | 0.360 | 0.495 | 0.455 | 0.588 | 0.591 | 0.679 |

Note: DISC = Discomfort; INSC = Insecurity; PU = Perceived Usefulness; PEU = Perceived Ease of Use; ATTI = Attitude; SN = Subjective Norms; BC = Perceived Behavioural Control; BI = Behavioural Intention; Trans = Transparency.

### 3.2. Hypothesis Testing

Table 5 presents a summary of the structural equation modelling results to examine the proposed relationships in this study. It shows the estimated latent variables' path coefficients with the corresponding t-statistics and significance values. All the tested model relationships were statistically significant ($p < 0.05$), with the independent constructs explaining 72.1% of the variation ($R^2 = 0.721$) with a strong correlation (r = 0.832). The results indicate that both discomfort and insecurity had a statistically significant negative effect on PEU, PU, PBC, and attitude towards the use of BCT when assessed at either a 1% or 5% significance level. The perceived ease of use (PEU), perceived usefulness (PU), perceived behavioural control (PBC), and attitude towards use (ATT) had positive influences on the behavioural intentions to use BCT in the Australian agricultural sector. The findings are shown in Table 5 and Figure 5.

**Table 5.** Hypothesis test results.

| Path | Path Coefficient | t-Statistics | Remarks |
| --- | --- | --- | --- |
| DISC → PEU | −0.503 *** | 6.245 | H1: Supported |
| DISC → PU | −0.091 *** | 5.891 | H2: Supported |
| DISC → PBC | −1.054 *** | 4.721 | H3: Supported |
| DISC → ATT | −0.435 *** | 5.012 | H4: Supported |
| INSC → PEU | −1.401 *** | 6.682 | H5: Supported |
| INSC → PU | −0.276 *** | 5.799 | H6: Supported |
| INSC → PBC | −0.030 *** | 4.549 | H7: Supported |
| INSC → ATT | −0.489 ** | 2.935 | H8: Supported |
| PEU → BI | 2.081 ** | 2.897 | H9: Supported |
| PU → BI | 2.341 ** | 2.902 | H10: Supported |
| PBC → BI | 1.891 *** | 5.564 | H11: Supported |
| ATT → BI | 1.544 ** | 2.967 | H12: Supported |

NB: *** $\rho < 0.001$ (significance at 0.1%); ** $\rho < 0.05$ (significance at 5%).

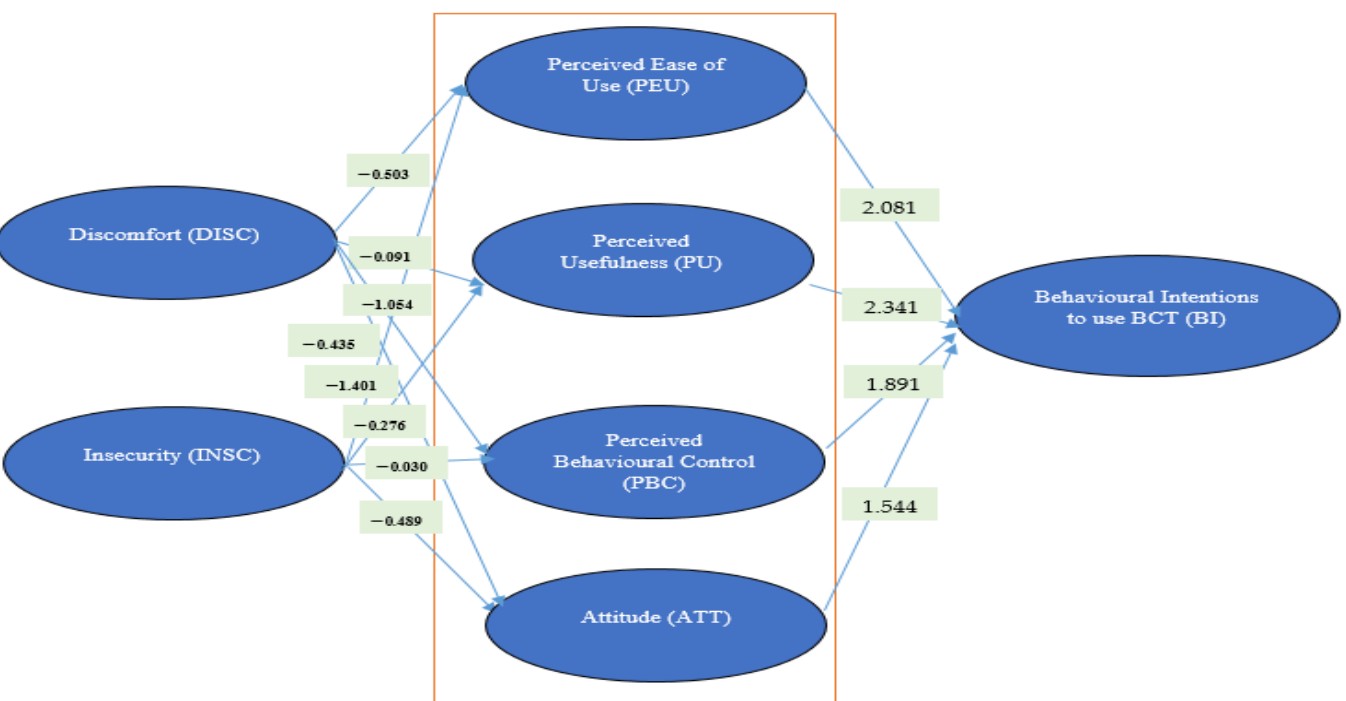

**Figure 5.** The results for the proposed model.

The PLS-SEM results are summarised as follows: Discomfort had a statistically significant negative effect on the perceived ease of use ($\beta = -0.503$, $p < 0.05$), perceived usefulness ($\beta = -0.091$, $p < 0.05$), perceived behavioural control ($\beta = -1.054$, $p < 0.05$), and attitudes ($\beta = -0.435$, $p < 0.05$), therefore leading to the acceptance of H1, H2, H3, and H4. Similarly, the PLS-SEM results show that the insecurity aspects of BCT had significant negative effects on the perceived ease of use ($\beta = -1.401$, $p < 0.05$), perceived usefulness ($\beta = -0.276$, $p < 0.05$), perceived behavioural control ($\beta = -0.030$, $p < 0.05$), and attitudes ($\beta = -0.489$, $p < 0.05$), indicating that H5, H6, H7, and H8 were supported based on the findings. Further, the perceived ease of use ($\beta = 2.081$, $p < 0.05$), perceived usefulness ($\beta = 2.341$, $p < 0.05$), perceived behavioural control ($\beta = 1.891$, $p < 0.05$), and attitudes ($\beta = 1.544$, $p < 0.05$) were found to have significant positive impacts on the behavioural intention to use/implement BCT in the Australian agricultural supply chain. This indicates that the PLS-SEM results support H9, H10, H11, and H12 when assessed at either a 1% or 5% significance level.

### 3.3. Post Hoc Analysis

The proposed hypotheses suggest that there is a mediation relationship with the perceived ease of use (PEU), perceived usefulness (PU), perceived behavioural control (PBC), and attitude as mediators. To examine the mediation effect, additional statistical bootstrapping testing was conducted. The mediation testing included all four mediator variables (PEU, PU, PBC, and attitude) given that they were found to be statistically significant. The results of the bias-corrected bootstrapping indicate that all the paths between each pair of variables included in the model were statistically significant. Specifically, the outcome of the bias-corrected bootstrapping summarised in Tables 6 and 7 indicate that the perceived ease of use (PEU), perceived usefulness (PU), perceived behavioural control (PBC), and attitude fully mediate the relationship between discomfort and the behavioural intentions to use BCT. Further, the bias-corrected bootstrapping testing results presented in Tables 6 and 7 show that all four mediator variables (PEU, PU, PBC, and attitude) fully mediate the relationship between insecurity and the behavioural intentions to adopt BCT.

**Table 6.** Results for mediation testing: perceived ease of use and perceived usefulness.

| Variables | Behavioural Intentions without PEU as Mediator | Behavioural Intentions with PEU as Mediator | Indirect Effects | Behavioural Intentions without PU as Mediator | Behavioural Intentions with PU as Mediator | Indirect Effects |
|---|---|---|---|---|---|---|
| Discomfort | −0.483 *** | −0.365 *** | −0.118 ** | −0.081 *** | −0.066 *** | −0.150 ** |
| Insecurity | −1.395 *** | −1.201 *** | −0.194 ** | −0.266 *** | −0.241 *** | −0.025 ** |

NB: *** $\rho < 0.001$ (significance at 0.1%); ** $\rho < 0.05$ (significance at 5%).

**Table 7.** Results for mediation testing: perceived behavioural control and attitude.

| Variables | Behavioural Intentions without PBC as Mediator | Behavioural Intentions with PBC as Mediator | Indirect Effects | Behavioural Intentions without Attitude as Mediator | Behavioural Intentions with Attitude as Mediator | Indirect Effects |
|---|---|---|---|---|---|---|
| Discomfort | −1.033 *** | −1.017 *** | −0.016 ** | −0.428 *** | −0.391 *** | −0.037 ** |
| Insecurity | −0.028 *** | −0.021 *** | −0.007 ** | −0.479 *** | −0.455 *** | −0.022 ** |

NB: *** $\rho < 0.001$ (significance at 0.1%); ** $\rho < 0.05$ (significance at 5%).

## 4. Discussion

The aim of this study is to identify the determinants of blockchain adoption in the agricultural supply chain. The obtained results show that the perceived behavioural control, perceived ease of use, perceived usefulness, attitude towards use, discomfort, and insecurity are related to blockchain technology adoption in agricultural supply chain management [2,53]. Consistent with the initial hypotheses, the discomfort and insecurity aspects of BCT were found to have an adverse effect on the perceived ease of use, perceived usefulness, perceived behavioural control, and attitude towards the use of blockchain technology in the agricultural sector. Furthermore, the findings based on structural equation modelling also indicated that the perceived ease of use, perceived usefulness, perceived behavioural control, and attitude towards the use of BCT had significant positive effects on stakeholders' behavioural intentions to use blockchain technology in the Australian agricultural sector.

These findings echo the observations from past studies, which reported that the discomfort and insecurity aspects of BCT tend to have negative effects on the perceived ease of use, perceived usefulness, and behavioural intentions to adopt blockchain technology in Indian supply chain management The findings are also fairly consistent with the observations from the Iranian and Malaysian supply chains that discomfort negatively impacts the behavioural intention to adopt blockchain technology [54,55].

### 4.1. Discomfort and BCT

Further insights indicate that discomfort negatively affects the perceived ease of use of blockchain technology adoption (H1), while it also has a negative impact on the perceived usefulness of BCT (H2). Discomfort associated with BCT was also found to have a negative effect on the stakeholders' perceived behavioural control (H3) and attitude towards the use of BCT (H4). These observations show that Australian supply chain specialists are likely to consider discomfort as a potential inhibiting factor in the blockchain technology adoption process. Concerns about discomfort may be attributed to uncertainties such as the lack of a universal ecosystem and platform for scaling up blockchain technology adoption and application [7,8]. These survey results confirm past findings that organisation managers who express extreme discomfort towards blockchain technology are less likely to adopt it in their supply chain management [40]. Similar observations were also reported in the United States and Italian food supply chains, where discomfort was documented to negatively influence the perceived usefulness of BCT, subsequently inhibiting its adoption [18,41].

### 4.2. Insecurity and BCT

The findings indicate that insecurity has a significant negative effect on both the perceived usefulness and perceived ease of use when adopting BCT in agricultural supply chains (confirming H5 and H6). Similarly, insecurity has a significant negative effect on stakeholders' perceived behavioural control and attitude towards the use of BCT (confirming H7 and H8). The implication based on the findings is that insecurity negatively affects blockchain technology uptake as organisational leaders become anxious and uncertain regarding its perceived usefulness and perceived ease of use [10,41]. These findings show that Australian supply chain specialists are likely to consider insecurity as a potential inhibiting factor in the blockchain technology adoption process [8].

### 4.3. BCT and Intentions to Use Blockchain Technology

The findings indicate that the perceived ease of use (H9), perceived usefulness (H10), perceived behavioural control (H11), and attitude towards the use of BCT (H12) have significant positive effects on the behavioural intentions to use blockchain technology in the Australian agricultural sector. The observations align with a growing body of the literature that shows the perceived ease of use to substantially impact managers' support and intentions to implement blockchain technology in agricultural supply chain management [8,44]. A primary impact of the perceived ease of use on blockchain technology adoption aligns with the extent that managers in agricultural organisations believe that using blockchain technology would improve aspects such as trust, transparency, and traceability in the supply chain process [7]. The survey findings confirm that the perceived ease of use is a major technology acceptance model determinant that informs BCT adoption in the agricultural sector. Further, perceived usefulness has been noted to influence blockchain technology adoption due to important attributes of blockchain such as its fast, effort-saving, timesaving, overall usefulness, and cost-saving qualities [43]. Both the perceived ease of use and perceived usefulness play important mediating roles in enhancing the relationship between discomfort and the behavioural intention to adopt BCT in the agricultural sector. Furthermore, these two variables (i.e., PEU and PU) were found to have significant mediating effects in explaining the relationship between insecurity and the behavioural intention to adopt BCT. The implication is that discomfort and insecurity affect the behavioural intention to adopt BCT through the perceived ease of use and perceived usefulness.

The construct of perceived behaviour control was found to positively affect the behavioural intentions to use blockchain technology in the agricultural industry (confirming H11). This study confirms previous findings that perceived behavioural control influenced organisational managers' intentions to use blockchain technology in their supply chain management [17,20]. The findings are consistent with past studies that have shown that perceived behavioural control tends to influence blockchain technology adoption among organisational leaders in their logistics management [1,2]. The mediation testing results

found that perceived behavioural control had a full mediating effect in enhancing the relationship between discomfort/insecurity and the behavioural intention to adopt BCT in the agricultural sector. The implication is that discomfort and insecurity have significant direct and indirect effects on the behavioural intentions towards BCT through the mediating role of perceived behavioural control.

The attitude towards blockchain technology adoption was also observed to positively affect the behavioural intentions to use blockchain technology in agricultural supply chains. The results further confirm H12 in that there is a positive impact between individual attitudes and the intention to use blockchain. These observations echo the findings from the literature on the potential enablers of blockchain technology adoption in supply chain management. For example, Wamba and Queiroz [17] reported that attitude defines an emotional aspect of managers' intentions to use blockchain technology. Positive emotions, beliefs, and behaviours about blockchain technology largely contribute to support for the adoption of the technology in supply chain systems [3,17]. The survey results show that having appropriate beliefs and emotions would impact practitioner support for BCT adoption, while a negative attitude would have a counterproductive impact on its uptake in supply chain management [9]. The mediation test findings indicate that the attitude towards technology plays a significant mediating role in enhancing the relationship between discomfort/insecurity and the behavioural intentions to adopt BCT in the agricultural sector.

## 5. Implications

### 5.1. Theoretical Implications

The results of this study have potential theoretical implications when considering the technology acceptance model, the theory of planned behaviour, and the technology readiness index constructs. This study contributes to the theory on supply chains by indicating how discomfort and insecurity influence the adoption of BCT in supply chains. Important constructs added to the theory include discomfort and insecurity as presented in the measurement items. The results show that both discomfort and insecurity negatively affect the perceived usefulness, perceived ease of use, perceived behavioural control, and attitude towards the use of BCT. These factors (i.e., PEU, PU, attitude, and PBC) play a mediating role in enhancing the relationship between discomfort/insecurity and the behavioural intentions to adopt BCT.

This study contributes to the body of knowledge on agriculture. Specifically, there is a paucity of studies that have examined this problem within agricultural supply chains. The insights from this study indicate that differences in culture, country, and ethnicity may influence the adoption of BCT. Managers in agricultural sectors need to evaluate the central roles that discomfort and insecurity may play in informing the uptake of BCT in agricultural sectors. The current study identifies important theoretical constructs that may be used to identify the drivers for the adoption of blockchain technologies in agricultural supply chains. Insights from this study identify the key technology constructs that may help us to understand the motivators that are likely to inform blockchain technology adoption.

The keys among the important theories of planned behaviour constructs that may influence the theory of planned behaviour adoption include subjective norms and their impact on perceived usefulness and perceived behaviour control. By contrast, the theory of planned behaviour constructs that impact blockchain technology adoption include discomfort and insecurity that impact the perceived ease of use and perceived usefulness. The technology acceptance model construct shows that the perceived ease of use and perceived usefulness largely impact the attitude towards blockchain technology adoption. Attitude also positively affects the behavioural intention to use blockchain technology.

### 5.2. Managerial and Practical Implications

The insights from this study have important implications for managerial practice within agricultural supply chain management. First, this study identified the essential

constructs for the successful adoption of BCT in Australian supply chain management. This study also showed how supply chain managers' behavioural intentions to adopt blockchain technology emerge based on behavioural control, individual attitudes, perceived usefulness, and perceived ease of use. Second, this study indicated that the theory of planned behaviour, the technology acceptance model, and the technology readiness index constructs were the keys to understanding managerial decisions when adopting and implementing blockchain technology in agricultural supply chains. Specifically, more focus on the successful adoption of blockchain should be anchored on addressing hurdles such as discomfort and insecurity since they might discourage managers from adopting blockchain technology in their organisations. The potential risks and concerns regarding BCT implementation in the agricultural supply chain include its insecurity, privacy issues, un-scalability, and problems in regulations [7]. This was confirmed by the negative path coefficients of discomfort and insecurity for each of the construct variables, perceived ease of use, perceived usefulness, perceived behavioural control, and attitudes. To address the BCT insecurity and privacy issues, managers of Australian agricultural firms will need to invest in training to manage the negative perceptions of BCT [1]. Further, ensuring that the BCT infrastructure and system complies with relevant standards and regulations could also help to address these concerns [14].

Third, while more managerial efforts should be taken to address the potential negative impacts of insecurity and discomfort on blockchain technology adoption, managers may optimise constructs including behavioural control, positive attitude, perceived usefulness, and perceived ease of use [1]. Based on the findings from the mediation testing, it was noted that incorporating these four constructs (i.e., PEU, PU, PBC, and attitude) would mitigate the adverse influence of discomfort/insecurity on the behavioural intention to adopt BCT. For instance, instilling a positive attitude among employees and managers with regard to BCT through training can be effective in mitigating the negative effect of discomfort/insecurity on the intentions to implement the technology within the organisation. Previous findings indicate that having positive perceptions of technology uptake and use potentially encourages organisations to implement blockchain technology in their logistics management systems [1]. These findings show that the perceived usefulness and perceived ease of use would enable stakeholders in the agricultural supply chain to perceive blockchain as being free of effort. As a result, organisations consider blockchain technology to enable them to derive maximum returns for their supply chain management.

The knowledge on blockchain adoption has three main practical implications and contributions. First, it provides a basis for enhancing business process capabilities through the integration of information systems and facilitating the completion of online transactions in a trustful environment. Second, it contributes to highlighting how the certainty aspect of blockchain due to its transparency and traceability has transformed digital marketing in the agricultural supply chain. These features of blockchain allow customers to track the source of advertised products. This highlights the importance of the "transparency" and "traceability" aspects of BCT in enhancing the efficiency and effectiveness of agricultural supply chains [2]. These attributes were noted to result in the improvement of the sustainable performance of agricultural supply chains [2,17]. The transparency and traceability features of BCT also facilitate the real-time monitoring of products across the supply chain, thereby ensuring that they meet the expected standards and regulatory requirements. The transparency and traceability features enable the proposed BCT-based agricultural supply chain to compare favourably to the current/conventional supply chain system. With the traditional non-BCT-based agricultural supply chain, managers find it difficult to trace the exact location and status of agricultural products in transit [2]. However, the proposed BCT-based agricultural supply chain allows for managers to track the status of their products in transit while being assured that they have been sourced and produced in a way that is consistent with the required standards and regulations [17]. Third, it contributes useful insight on how BCT can improve farming experience and productivity by creating connectivity with customers and farm input suppliers.

To effectively implement BCT in the Australian agricultural sector, there are certain ethical and legal aspects that should be considered by managers. The technology has a potential to interfere with data confidentiality, thereby creating data privacy concerns in corporate entities [2]. Further, managers need to be aware that in most countries, there is lack of effective data governance regulations/laws that would address data access and control issues. These legal challenges can be effectively addressed when relevant legislative authorities enact laws that manage the collection, sharing, and use of BCT-based data in agricultural supply chains [56–58]. Training could also be important in helping to address the ethical and privacy concerns of BCT adoption in the Australian agricultural sector [2].

## 6. Limitations and Future Research

A key limitation of this study was that most information was based on numerical data that were collected using surveys. Future studies should be undertaken using field observations, interviews, and focus group discussions to collect participants' feelings, lived experiences, and personal attitudes towards blockchain technology. In addition, company data such as archival information and minutes of board meetings among various agricultural organisations may help us to understand the commitment that managers and other stakeholders have towards blockchain adoption in their supply chain management.

**Author Contributions:** Conceptualization, P.S.G.M.; methodology, P.S.G.M.; software, P.S.G.M.; validation, P.S.G.M.; formal analysis, P.S.G.M.; investigation, C.P. and A.-M.S.; resources, P.S.G.M.; data curation, P.S.G.M.; writing—original draft preparation, P.S.G.M.; writing—review and editing, P.S.G.M. and A.-M.S.; visualization, P.S.G.M.; supervision, A.-M.S., J.S. and C.P.; project administration, P.S.G.M., J.S. and A.-M.S.; funding acquisition, NA. All authors have read and agreed to the published version of the manuscript.

**Funding:** This research received no external funding.

**Data Availability Statement:** The data that support the findings of this study are available upon request from the corresponding authors.

**Conflicts of Interest:** The authors declare that there are no conflict of interest involved in this study.

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
