# Peer review of "Identifying an Optimal Model for Blockchain Technology Adoption in the Agricultural Sector"

_logistics_

Round 1
Reviewer 1 Report
The main objective is not reflecting
Motivation is very weak in the introduction section.
All the hypotheses are validated through a theoretical perspective. None of the hypotheses were validated through simulation . The authors are presented in Table 5. Hypothesis test results but not convinced. Required more elaborate on it .
Some major flaws
- How important is transparency and traceability in your agricultural supply chain, and how do you ensure that your products meet the necessary standards and regulations?
- Authors are suggested to mention the potential risks and concerns that you have about implementing blockchain technology in your agricultural operations, and how those can be addressed. I tried to find this answer but couldn't
- How do you currently manage and track the provenance of your agricultural products, and what are the limitations of your current system? The authors have not compared at the existing system to the proposed system.(I haven't seen any where SOTA approach has been used.)
- What are the ethical and legal considerations that need to be taken into account when implementing blockchain technology in agriculture, and how can they be addressed?
Ok
Author Response
Dear Sir/Madam,
We highly value the valuable and insightful feedback provided by you, which has greatly contributed to enhancing the manuscript. Your comments were thoroughly taken into account during the process of revising the manuscript. We are now submitting the revised version of the manuscript, incorporating the suggestions from the reviewers. Kindly find the attached file containing our responses to the comments.
Thank you for your time and consideration.
Kind Regards,
Peter Gandhi

Reviewer 2 Report
Congratulations for an interesting topic which gathers Agriculture, Social sciences and Technology under one consistent approach!
I have a couple of questions if you're kind to respond:
1. How do the farmers respond to BCT in development countries?
2. Have you consider separate studies on each level of the supply chain?
3. Do you consider TAM model appropriate for the Agricultural sector?
4. Does BCT integrates well with the biological cycle of the plants and animals?
5. Is it possible to estimate results by differentiating the study on bio/eco sustainable agriculture and industrialized agriculture?
Thank you!
Author Response

(The authors gave the same response as above.)

Round 2
Reviewer 1 Report
Authors are suggested to check grammatical corrections,Quality of English Language .
ok